# Generative Neural Articulated Radiance Fields

**Alexander W. Bergman**\*
Stanford University
awb@stanford.edu

**Petr Kellnhofer**\*
TU Delft
p.kellnhofer@tudelft.nl

**Wang Yifan**\*
Stanford University
yifan.wang@stanford.edu

**Eric R. Chan**\*
Stanford University
erchan@stanford.edu

**David B. Lindell**
University of Toronto
Vector Institute
lindell@cs.toronto.edu

**Gordon Wetzstein**
Stanford University
gordonwz@stanford.edu

computationalimaging.org/publications/gnarf/

## Abstract

Unsupervised learning of 3D-aware generative adversarial networks (GANs) using only collections of single-view 2D photographs has very recently made much progress. These 3D GANs, however, have not been demonstrated for human bodies and the generated radiance fields of existing frameworks are not directly editable, limiting their applicability in downstream tasks. We propose a solution to these challenges by developing a 3D GAN framework that learns to generate radiance fields of human bodies or faces in a canonical pose and warp them using an explicit deformation field into a desired body pose or facial expression. Using our framework, we demonstrate the first high-quality radiance field generation results for human bodies. Moreover, we show that our deformation-aware training procedure significantly improves the quality of generated bodies or faces when editing their poses or facial expressions compared to a 3D GAN that is not trained with explicit deformations.

## 1 Introduction

Unsupervised learning of 3D-aware generative adversarial networks (GANs) using large-scale datasets of unstructured single-view images is an emerging research area. Such 3D GANs have recently been demonstrated to enable photorealistic and multi-view consistent generation of radiance fields representing human faces [1–7]. These approaches, however, have not been shown to work with human bodies, partly because learning the body pose distribution is much more challenging given the significantly higher diversity in articulations compared to facial expressions.

Yet, generative 3D models of photorealistic humans have significant utility in a wide range of applications, including visual effects, computer vision, and virtual or augmented reality. In these scenarios, it is critical that the generated people are editable to support interactive applications, which is not necessarily the case for existing 3D GANs. While variations of linear blend skinning [8] have been adopted to articulate radiance fields for single-scene scenarios [9–21], it is unclear how to efficiently apply such deformation methods to generative models.

With our work, dubbed generative neural articulated radiance fields or GNARF, we propose solutions to both of these challenges. Firstly, we demonstrate generation of high-quality 3D (i.e., multi-view consistent and geometry aware) human bodies using a GAN that is trained in an unsupervised manner on datasets containing single-view images. To this end, we adopt the recently proposed tri-plane

---

\*Equal contribution

36th Conference on Neural Information Processing Systems (NeurIPS 2022).

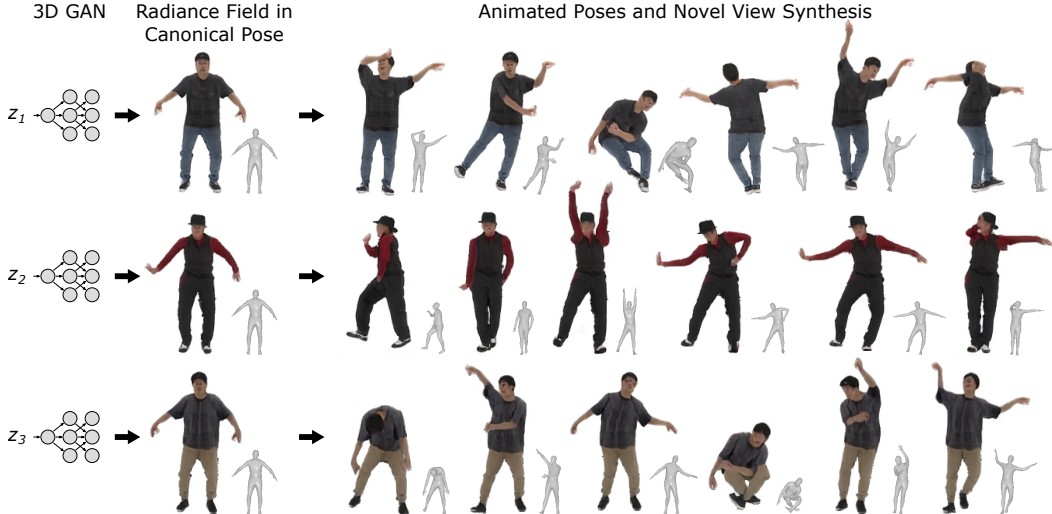

Figure 1: Our method, GNARF, maps a latent space to radiance fields representing human identities. These generated humans can then be animated and rendered from novel views. Qualitative results for our method trained on the AIST++ dataset [23] are shown here.

feature representation [1], which is extremely efficient for training and rendering radiance fields, while being compatible with conventional 2D CNN–based generators, such as StyleGAN [22]. While this framework has been successfully demonstrated for faces in prior work, we are the first to adapt it to generating radiance fields of full human bodies. Secondly, we tackle the editability of the generated radiance fields by introducing an explicit radiance field deformation step as part of our GAN training procedure. This step ensures that the generator synthesizes radiance fields of people in a canonical body pose, which is then explicitly warped according to the body pose distribution of the training data. We show that this new approach generates high-quality, editable, multi-view-consistent human bodies and that our approach can also be applied to editing faces, increasing the controllability of existing generative models for this task (see Fig. 1).

To summarize, the contributions of our approach are:

- We present a 3D-aware GAN framework for the generation of editable radiance fields of human bodies. To our knowledge, this is the first approach of its kind.
- Our framework introduces an efficient neural representation for articulated objects, including bodies and heads, that combines the recently proposed tri-plane feature volume representation with an explicit feature volume deformation that is guided by a template shape.
- We demonstrate high-quality results for unconditional generation and animation of human bodies using the SURREAL and AIST++ datasets and faces using the FFHQ dataset.

## 2 Related Work

**Articulated 3D Representations.** Parametric shape templates are one of the most common types of articulated 3D representations adopted in recent neural scene representation and rendering approaches. These templates, including faces [24, 25], bodies [26], hands [27], or a combination of these parts [28], and even animals [29], have been widely utilized for pose estimation and reconstruction, e.g. [30–34]. Volume deformation is also commonly used in computer graphics, for example using mean value coordinates (MVC) [35] or biharmonic coordinates [36] for shape deformation and editing [37–41]. GNARF combines parametric template shapes, such as FLAME [25] for heads and SMPL [26] for bodies, with an intuitive surface-driven volume deformation approach that is both computationally efficient and qualitatively comparable to or better than both skinning and MVC-based deformation. This provides intuitive editing control for articulated radiance field deformation.

**Neural Radiance Fields.** Coordinate networks, also known as neural fields [42], have emerged as a powerful tool that enable differentiable representations of 3D scenes [43–56] and learning view-dependent neural radiance fields [57–83]. While initial proposals have focused on static scenarios,

recent work has demonstrated successful representations of dynamic scenes [84–91]. Articulated neural radiance fields further extend these approaches by providing editability for neural representations of human heads [92–97] and bodies [9–21, 98, 99] or animals [100], often by deforming the underlying radiance fields using traditional 3D morphable models or skeleton-based parameterizations, or alternatively conditioning the radiance field decoder with pose-related parameters. A more detailed survey of static, dynamic, and articulated neural radiance fields can be found in the recent state-of-the-art report by Tewari et al. [101]. Note that all of these techniques are supervised with scene-specific multi-image data and focus on representing, i.e., "overfitting", a single scene. Therefore, it is not easily possible to train these models using unstructured 2D image data and then apply them to generate and edit new and unseen objects or humans.

**Generative 3D-aware Radiance Fields.** Building on the success of 2D image-based GANs [22, 102–104], recent efforts have focused on training 3D-aware multi-view consistent GANs from collections of single-view 2D images in an unsupervised manner. Achieving this challenging goal requires a combination of a neural scene representation and differentiable rendering algorithm. Recent work in this domain builds on representations using meshes [105, 106], dense [107–112] or sparse [113] voxel grids, multiple planes [2], fully implicit networks [3–7], or a combination of low-resolution voxel grids combined with 2D CNN-based image upsampling layers [114, 115]. Our 3D GAN architecture is most closely related to the recent work by Chan et al. [1], which uses an efficient tri-plane-based volume representation combined with neural volume rendering. We extend this work by including an explicit deformation field in our GAN architecture to model diverse articulations, which allows the generator to synthesize radiance fields of human bodies or heads in a canonical pose while being supervised by 2D image collections that contain arbitrary pose distributions. Explicitly disentangling radiance field generation and deformation enables us to drastically improve the quality of generated human bodies and faces when their poses or facial expressions are edited.

HeadNeRF [116] is related to our approach in that they generate 3D heads conditioned on various attributes. Both HeadNeRF and GNARF condition on identity and facial expression independently, with the former also conditioning on illumination and albedo. However, to disentangle individual attributes they need to acquire training images of the same person performing various expressions in different lighting conditions. In contrast, and similar to other 3D GANs, our approach only requires single-view images of different people and can therefore work with readily available 2D image collections. The recent work by Grigorev et al. [117] also generates human bodies. However, their work proposes a 2D GAN that generates textures which are used in combination with a standard articulated template mesh whereas we aim at generating and editing radiance fields using a 3D GAN. The concurrent work of Noguchi et al. [118] is closest to ours as the method also includes a radiance field deformation step in a tri-plane-based 3D GAN. Our evaluation shows superior generation quality and, more importantly, while their approach ties the network architecture to the specific choice of skeleton, our surface-driven volume deformation is agnostic to the particular choice of template and can be used with human bodies, faces, or other object types.

## 3 Generative Articulated Neural Radiance Fields

GNARF is a novel general framework to train 3D-aware GANs for deformable objects that have a parametric template mesh, e.g. human bodies and faces. It builds on the efficient tri-plane feature representation [1] for the generated neural radiance field, but additionally applies an explicit deformation which alleviates the requirement for the generator to learn a complicated distribution of articulations. As a result, the generator automatically learns to generate radiance fields of objects in the canonical pose, which are then warped explicitly to produce target body poses and facial expressions in a fully controllable and interpretable manner.

### 3.1 Modeling Articulated Radiance Fields

We first discuss our approach to modeling and rendering articulated radiance fields, before describing how this is integrated into the 3D GAN in Sec. 3.2.

**Scene representation.** To represent an object, we leverage the recently proposed tri-plane feature representation [1]. This representation uses three axis-aligned 2D feature planes, each with resolution $N \times N \times C$, where $N$ and $C$ denote the spatial resolution and number of channels. The feature of any

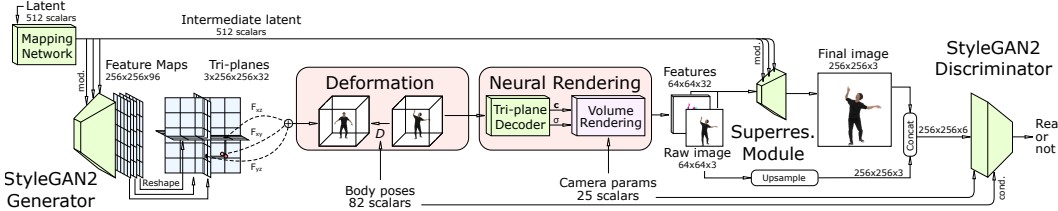

Figure 2: Illustration of the GNARF pipeline, including the StyleGAN2 generator, the tri-plane feature representation, feature volume deformation, neural volume rendering, image super-resolution as well as camera view and body-pose conditioned dual discrimination. The resolution of intermediate data and the final image is indicated for experiments with the AIST++ dataset.

3D point $\mathbf{x} \in \mathbb{R}^3$ is queried by projecting $\mathbf{x}$ onto the planes, retrieving three feature vectors via bilinear interpolation, and aggregating the vectors by summation, i.e., $F(\mathbf{x}) = F_{xy}(\mathbf{x}) + F_{yz}(\mathbf{x}) + F_{xz}(\mathbf{x})$ where $F_{ij} : \mathbb{R}^3 \mapsto \mathbb{R}^C$ is a function mapping 3D coordinates to features on the $ij$ plane via projection and interpolation.

**Articulated deformation.** We use the deformation function $D : \mathbb{R}^3 \mapsto \mathbb{R}^3$ (detailed later) to warp a coordinate $\mathbf{x}$ from the target (deformed) space into the canonical space. Using a small multilayer perceptron $\text{MLP} : \mathbb{R}^C \mapsto \mathbb{R}^4$, we convert the deformed 3D feature volume into a neural field of spatially varying RGB colors $\mathbf{c}$ and volumetric densities $\sigma$ as

$$(\mathbf{c}(\mathbf{x}), \sigma(\mathbf{x})) = \text{MLP}\left((F_{xy} \circ D)(\mathbf{x}) + (F_{yz} \circ D)(\mathbf{x}) + (F_{xz} \circ D)(\mathbf{x})\right). \tag{1}$$

There are many possible choices for how to specify the deformation field in an intuitive manner. For example, linear blend skinning can be used to deform the entire volume "rigged" by a skeleton (see e.g. [8]). While skinning is popular for human body articulations, it cannot explain subtle deformation due to varying facial expressions. Another option is to use the object-specific template mesh as a cage and apply cage-based deformation for the entire volume using mean value coordinates (MVC) [35]. However, the high computational cost of evaluating MVCs on the full-resolution grid (see Tab. 1) is prohibitive for GAN training and, more critically, this approach generally leads to severe artifacts when the template mesh (accidentally) includes self-intersections (see supplement).

To alleviate these problems, we use an intuitive surface-driven deformation method, which we label as the Surface Field (SF) method. This method only requires canonical and target template meshes with correspondences, which are readily available for faces [25] and bodies [26]. These template shapes, in turn, can be driven using skeletons, manual editing, or using keypoints or landmarks that could be detected in and transfered from videos of other people. Therefore, the SF method is generally sufficient to apply to different body parts and it can be intuitively edited in a number of ways, resulting in accurate volume deformation for our class of volumetric models.

The SF approach assigns each 3D coordinate $\mathbf{x}$ to its nearest triangle $t_{\mathbf{x}}^{\mathrm{D}} = [\mathbf{v}_0, \mathbf{v}_1, \mathbf{v}_2] \in \mathbb{R}^{3 \times 3}$ on the target (deformed) mesh. We compute the barycentric coordinates $[u, v, w]$ of the coordinate projected onto this triangle and find the corresponding triangle on the canonical mesh $t_{\mathbf{x}}^{\mathrm{C}}$ and its normal $\mathbf{n}_{t_{\mathbf{x}}}^{\mathrm{C}}$. The deformed coordinate can then be computed as

$$D(\mathbf{x}) = t_{\mathbf{x}}^{\mathrm{C}} \cdot [u, v, w]^{\mathsf{T}} + \left\langle \mathbf{x} - t_{\mathbf{x}}^{\mathrm{D}} \cdot [u, v, w]^{\mathsf{T}}, \mathbf{n}_{t_{\mathbf{x}}}^{\mathrm{D}} \right\rangle \mathbf{n}_{t_{\mathbf{x}}}^{\mathrm{C}}, \tag{2}$$

The SF approach is very fast to compute and mitigates artifacts from self-intersections of the template shape, thereby combining the benefits of linear blend skinning and MVC-based approaches for the task of radiance field deformation.

**Rendering deformed radiance fields.** We render the radiance field using (neural) volume rendering [62, 119]. For this purpose, the aggregated feature $F(\mathbf{r})$ of a ray $\mathbf{r}$ is computed by integrating the volumetric features $\mathbf{f}$ and density $\sigma$ as

$$\mathbf{F}(\mathbf{r}) = \int_{t_n}^{t_f} T(t)\, \sigma(\mathbf{r}(t))\, \mathbf{f}(\mathbf{r}(t))\, \mathrm{d}t, \quad T(t) = \exp\left(-\int_{t_n}^{t_f} \sigma(\mathbf{r}(s))\, \mathrm{d}s\right), \tag{3}$$

where $t_n$ and $t_f$ indicate near and far bounds along the ray $\mathbf{r}(t) = \mathbf{o} + t\mathbf{d}$ pointing from its origin $\mathbf{o}$ into direction $\mathbf{d}$. The volume rendering equation (eq. 3) is typically approximated using numerical methods, such as the quadrature rule [119].

## 3.2 3D GAN Framework

An overview of our 3D pipeline is shown in Fig. 2. Several components, including the StyleGAN generator, the tri-plane representation, the volume rendering, the CNN-based image super-resolution module, and (dual) discrimination are directly adopted from the EG3D framework [1].

Instead of directly generating the radiance field with the target body pose or facial expression, however, GNARF is unique in generating the radiance field in a canonical pose and then applying the deformation field discussed in the previous section to warp the feature volume. We additionally remove the pose conditioning on the generator, and only use camera pose and body pose conditioning in the discriminator. This removes the ability for the generator to incorporate any knowledge about the final view or pose in the canonical radiance field generation, ensuring that the generated results will be robustly animatable beyond just the image rendered at training time. Thus, the generator depends only on the latent code controlling identity, which is input into a StyleGAN2 generator. This architectural choice takes advantage of the state-of-the-art 2D generative model architectures by using them to generate the tri-plane 3D representation. The discriminator having access to the camera and body poses ensure that the GAN learns to generate warping accurate to a target pose rather than just being in the correct distribution. Finally, we adopt a radiance field rendering strategy which samples along each ray inside of an expanded template mesh. This ensures that the integration samples are taken in regions of the radiance field with the most detail and not taken in empty space, simultaneously improving the quality of the generated results and speeding up training.

Additional implementation details, source code, and pre-trained models can be found in the supplement or on our website.

# 4 Experiments

We first evaluate the proposed deformation field by overfitting a single representation on a single dynamic full body scene. Then we apply this deformation method in a GAN training pipeline for both bodies (AIST++ [23] and SURREAL [120]) and faces (FFHQ [104]). Training details and hyper-parameters are discussed in the supplement.

AIST++ is a large dataset consisting of 10.1M images capturing 30 performers in dance motion. Each frame is annotated with a ground truth camera and fitted SMPL body model. SURREAL contains 6M images of synthetic humans created using SMPL body models in various poses rendered in indoor scenes. FFHQ is a large dataset of high-resolution images of human faces collected from Flickr. All images have licenses that allow free use, redistribution, and adaptation for non-commencial use.

## 4.1 Single-scene Overfitting

We compare the proposed surface-driven deformation method, SF, with two alternative methods, MVC and skinning, in a single-scene overfitting task. MVCs require a set of weights (called the mean value coordinates) to be computed w.r.t. every vertex of the target mesh $\mathcal{M}^D$ for every sample point. The sample point is then deformed into the canonical pose by linearly combining the vertices of the canonical mesh $\mathcal{M}^C$ with these computed weights. In skinning, the sampling points are deformed to the canonical pose by the rigid transformation of the closest bone as measured by point to line-segment distance. We find this simplified definition of skinning effective in avoiding blending between two topologically distant body parts (e.g., hand and pelvis) if the starting pose brings them to a geometric proximity.

We select a multi-view video sequence from the AIST++ dataset [23] and optimize tri-plane features in the canonical pose using a subset of the views and frames for supervision. We then evaluate the quality of the estimated radiance field warped into these training views and poses but also into held-out test views and poses. We apply several modifications to the tri-plane architecture to reduce overfitting; details regarding these changes as well as the selection of training and evaluation views are provided in the supplemental material.

|  | Training images | | | Test images | | | Run time [ms] ↓ |
|---|---|---|---|---|---|---|---|
|  | PSNR ↑ | SSIM ↑ | LPIPS ↓ | PSNR ↑ | SSIM ↑ | LPIPS ↓ |  |
| Skinning | 18.8 | 0.942 | 0.060 | 17.9 | **0.940** | 0.067 | 95.6 |
| MVC [35] | 18.1 | 0.937 | 0.067 | 17.2 | 0.934 | 0.074 | 0.2 (3782.1) |
| Surface Field | **19.0** | **0.943** | **0.058** | **18.0** | **0.940** | **0.065** | **31.6** |

Table 1: Single-scene overfitting. We evaluate three deformation approaches for the task of estimating a single radiance field in a canonical body pose supervised by a video sequence showing a person from different views and in different poses. The SF approach achieves the best quality for both training and unseen test images while being the fastest. Note that the MVC method is only faster than SF when using precomputed grid at the cost of significantly lower deformation accuracy (the runtime without such approximation is reported in parentheses). The timings are measured to deform a single feature volume on an RTX3090 graphics processing unit.

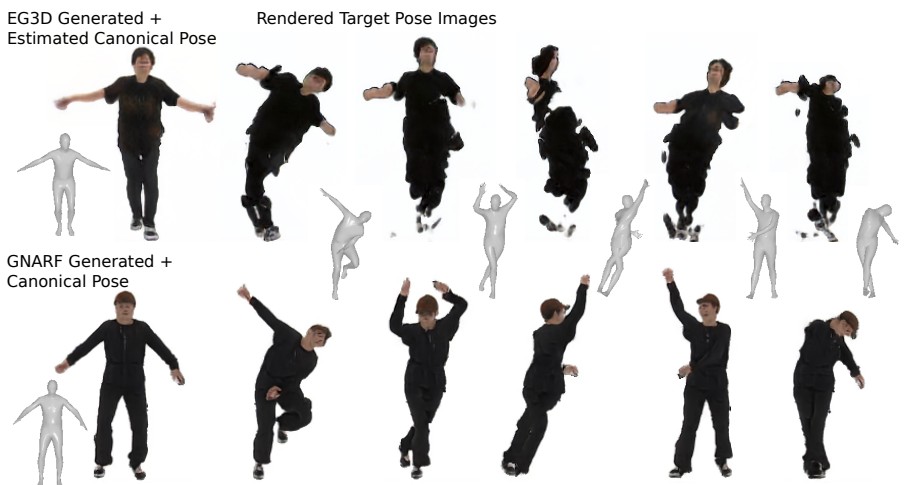

Figure 3: Qualitative comparison of generated target poses using our model vs. warping a pre-trained EG3D model on the AIST++ dataset.

To speed up MVC and SF computation, we decimate the source and deformed SMPL mesh using Quadric Error Metric Decimation [121] in the Open3D library [122] from the original 13,776 faces to 1,376 faces, while tracking the correspondence between the source and deformed meshes. Nonetheless computing MVC for each deformed pose is still prohibitively expensive for online training (3.7 s per example). We thus precompute the deformation for training and test body poses on a fixed $16^3$ grid and retrieve the deformation for arbitrary sampling points using trilinear interpolation.

As shown in Tab. 1, our SF method outperforms the others for both training and test images. MVC performs worst, partially due to the grid approximation, which is essential in practice. The skinning method is comparable to SF in terms of image quality but it is $3\times$ slower. Moreover, skinning cannot sufficiently deform subtle facial expressions. Therefore, the SF approach is the most flexible among these deformation methods by being compatible with different human body parts while also offering computational and memory efficiency.

## 4.2 Human Body Generation and Animation

We now use our SF approach as the deformation method for feature volumes generated by GNARF. Our method is trained and evaluated on the captured AIST++ [23] and the synthetic SURREAL [120] datasets. For both datasets, our method generates high-quality multi-view consistent human bodies in diverse poses that closely match the target pose.

**Baselines & Evaluation.** Because GNARF is the first method to learn a generative model of radiance fields representing bodies, we propose a baseline where we use the original EG3D trained without deformation to generate a feature volume (not in the canonical pose) then warp it into various target poses during inference time using the proposed SF deformation method. Without the feature

| | AIST++ @$256^2$ | | SURREAL @$128^2$ | | |
| --- | --- | --- | --- | --- | --- |
| | FID (50k) ↓ | PCKh@0.5 ↑ | FID (10k) ↓ | FID (50k) ↓ | PCKh@0.5 ↑ |
| ENARF-VAE [118]* | — | — | 63.0 | — | — |
| ENARF-GAN [118]* | — | — | 21.3 | — | 0.966 |
| EG3D (no warping) | 8.3 | — | 14.3 | 13.3 | — |
| EG3D (+ pose est. & re-warping) | 66.5 | 0.855 | 163.9 | 162.2 | 0.348 |
| GNARF | **7.9** | **0.980** | **4.7** | **5.7** | **0.999** |

Table 2: Quantitative evaluation of our GAN trained on SURREAL [120] and AIST++ [23]. Our method only considers foreground images (backgrounds masked). *Metrics provided by authors of [118] after standardizing the evaluation protocol, which may differ from initial values in their original report.

volume deformation, the generator is forced to learn to model both identity and pose in its latent space. As a result, the tri-plane features no longer represent a human body in a consistent canonical pose, but rather match the distribution of poses in the dataset. The animation of generated bodies is similar to our proposed approach, except that the generated (arbitrarily posed) human body is used as the canonical pose, for which we obtain a SMPL mesh by applying the human shape reconstruction method SPIN [31]. Additionally, we included the concurrent work ENARF-GAN and its variant ENARF-VAE [118] in the evaluation whenever a fair and standardized comparison is possible.

We compare FID scores to evaluate the quality and diversity of generated images as well as the Percentage of Correct Keypoints (PCK) metric to evaluate the quality of the animation. PCK computes the percentage of 2D keypoints detected on a generated rendered image that are within an error threshold (half the size of the head in the case of PCKh@0.5) of keypoints on a ground truth image in the same pose and view. We use an off-the-shelf body keypoint estimator [123] trained on MPII [124] publically available on the MMPose Project [125] to estimate keypoints from a corresponding ground truth and generated image.

**AIST++.** AIST++ is a challenging dataset as the body poses are extremely diverse. We collect 30 frames per video as our training data after filtering out frames whose camera distance is above a threshold or the human bounding box is partially outside the image. Then we extract the human body by cropping a $600 \times 600$ patch centered at the pelvis joint, and resize these frames to $256 \times 256$. Since this dataset does not provide ground truth segmentation masks, we use a pre-trained segmentation model [126] to remove backgrounds to stabilize the GAN training. To speed up training, we use GAN transfer learning [127]. Rather than initializing our network weights randomly, we begin training from a pre-trained EG3D [1] model. Fine-tuning allows for quicker convergence and saves computational resources during training.

As shown in Tab. 2, we can see that our method outperforms the naive reanimation of EG3D by a large margin (see the 4th row vs. the 5th row). Furthermore, our animated method also generates better images than the EG3D baseline that does not support animations. This is likely due to GNARF allowing the generator to focus on generating a specific identity in a canonical pose instead of learning both identity and complex pose distribution in a combined latent space. Similarly, our model enables high-quality re-animation, as demonstrated by the PCKh@0.5 metric. In Fig. 3, our method produces significantly better qualitative results than those generated by the baseline. The baseline results using re-warped EG3D are significantly degraded, since it is difficult to accurately estimate the SMPL mesh from the generated images, which we use as the canonical pose in the deformation function. Additionally, floating artifacts which exist in the radiance field outside of camera views and make no difference in the conventionally rendered images become visible after being warped

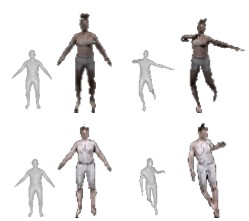

Figure 4: Example of generated humans in canonical pose and in target pose using model trained on SURREAL dataset.

can cause false occlusions. In Fig. 1, we show that our method generates bodies in a canonical pose with diverse identities. Additionally, we show that by changing the SMPL parameters, we can drive each radiance field to a desired target pose and render at an arbitrary novel view.

**SURREAL.** We also test our method on the SURREAL dataset. The training data is extracted from the first frame of each video in SURREAL's training split. Each frame is cropped based

|  | FID (500) ↓ | FID (50k) ↓ | AED ↓ | APD ↓ | ID-Consistency ↑ |
|---|---|---|---|---|---|
| EG3D (+3DMM est. & re-warping) | 22.9 | 11.6 | 0.29 | 0.028 | **0.81** |
| PIRenderer [129] | 64.4 | — | 0.28 | 0.040 | 0.70 |
| 3D GAN inversion [130] | 31.2 | — | 0.36 | 0.039 | 0.73 |
| GNARF | **17.9** | **6.6** | **0.23** | **0.025** | 0.80 |

Table 3: Quantitative comparison on FFHQ dataset [104].

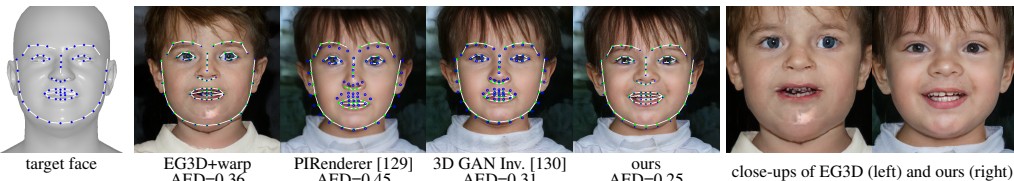

| target face | EG3D+warp AED=0.36 | PIRenderer [129] AED=0.45 | 3D GAN Inv. [130] AED=0.31 | ours AED=0.25 | close-ups of EG3D (left) and ours (right) |

Figure 5: Deformation fidelity on FFHQ dataset. We plot the target and detected facial landmarks in blue and green, respectively, to visualize the expression fidelity. PIRender [129] and 3D GAN inversion [130] both show large discrepancy to the target face, while the baseline EG3D has strong artifacts from warping (right). Our generated results show high visual quality and they align accurately with the driving deformation.

on the provided segmentation mask from head-to-toe, and resized to $128 \times 128$. We filter out the backgrounds and set them to be black. The SURREAL dataset provides ground truth SMPL parameters and camera intrinsics and extrinsics for each frame. Similarly to AIST++, we use transfer learning from a pre-trained EG3D model at the appropriate resolution.

In Tab. 2, we see that our method trained with deformation produces improved FID scores as the version trained without SF feature volume deformation (5th vs 3rd row). Attempting to deform generated radiance fields results in an immense degradation in quality, shown by both the FID and PCKh@0.5 metrics (4th row).

Additional qualitative results and evaluations are included on the website.

### 4.3 Human Face Generation and Editing

Thanks to the surface-driven deformation method, SF, our method can directly utilize expressive parametric face models to drive subtle deformations. We use the FLAME [25] head model and apply the state-of-the-art facial reconstruction method DECA [128] to estimate the flame parameters (100D identity vectors, 6D joint position vectors and 50D expression vectors) from the training dataset. Since DECA does not account for eyeball movement, we remove the eyeballs from the original FLAME template. Additionally, we add triangles to close the holes at the neck and mouth area in the FLAME template, which improves the consistency of the SF deformation. These preprocessing steps produce a mesh with 3,741 vertices and 7,478 faces. Finally we apply the same decimation method as in the body experiments to obtain a coarse mesh (2,500 triangles), which we use during the GAN training for faster SF deformation.

To train GNARF, we start with a conventional EG3D model pre-trained using the FFHQ dataset (see [1] for details on the training details and data processing). We then fine-tune this model using the proposed framework, which includes the SF deformation module. Note that we apply a global scaling and translation to the FLAME template mesh to roughly align it to the faces generated by the pre-trained EG3D model.

For evaluation, our first baseline is the original EG3D model with re-warping at inference time, as described in Sec. 4.2. We also include two state-of-the-art facial reenactment methods, PIRenderer [129] and 3D GAN inversion [130]. Several metrics are used by our quantitative evaluation. FID (500) follows the evaluation protocol of Lin et al. [130], where the ground truth dataset consists of 500 randomly sampled identities and the test dataset is constructed by animating the ground truth using randomly sampled target poses. FID (50k) follows the protocol in EG3D, where the entire FFHQ dataset is treated as the ground truth and the test dataset includes 50k generated images using randomly sampled latent vectors, camera poses and FLAME facial parameters. Following [130], we evaluate the fidelity of the animation with the Average Expression Distance (AED) and the Average

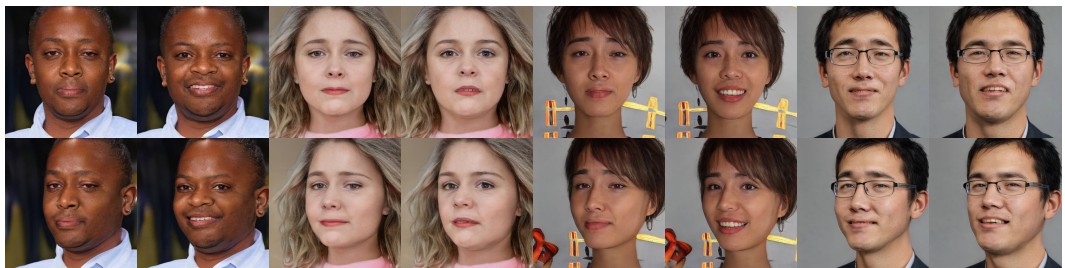

Figure 6: Qualitative results on FFHQ dataset. We show each identity in their canonical (left) and a different (right) expression from two different camera poses. Our method shows excellent multi-view consistency.

Pose Distance (APD) computed using DECA estimation, as well as the identity consistency based on a face recognition model [131].

As shown in Tab. 3, our method is superior to the baseline methods in both FIDs and in the edited fidelity, and comparable to the baseline EG3D in terms of identity consistency. The advantage of our method in editing ability is clearly demonstrated in Fig. 5: our method reproduces the target expression more accurately when compared to the face reenactment methods and mitigates the warping artifacts present in the baseline EG3D. In Fig. 6, we further demonstrate the quality of our results. Notice that even though the FLAME model does not include teeth, the SF deformation guides the neural radiance field to construct teeth consistently at the correct location.

## 5    Discussion

**Limitations and future work.**    Our work is not without limitations.   The level of detail in the generated bodies, for example, is relatively low. This is partly due to the limited resolution of the training data in the SURREAL and AIST++ datasets, but also due to the limited resolution that the tri-plane representation offers for any one body part, such as the face. An interesting avenue of future work could include the exploration of adaptive radiance field resolution for human bodies, allocating more resolution to salient parts (see e.g. [132]). The image quality of the generated bodies is currently on par with simpler approaches that only need to generate an RGB texture on the SMPL mesh [117]. Yet, details in faces and hair cannot be handled by a texture-generating approach and we expect the quality of generative radiance fields to surpass that of simpler alternatives with increasing and perhaps adaptive radiance field resolutions. Another bottleneck of our current framework is the challenge of being able to work with large-scale datasets showing a diversity of visible humans. In-the-wild datasets, such as MSCOCO [133], do contain this diversity but also contain a significant amount of occlusion and detailed backgrounds, requiring the generator to spend capacity on modeling the distribution of backgrounds and occlusions themselves. An interesting avenue of future work consists of explicitly modeling occlusion and background in the GAN training pipeline independently from identity and pose. Currently, the background is not modeled in the generative framework and requires pre-processing of the dataset to separate foreground from background. Additionally, pose estimation from in-the-wild images is still not very accurate, degrading the quality of our deformation function. On the other hand, large-scale custom curated datasets such as the one used in InsetGAN [132] are not publicly available. Finally, the deformation we use as part of the 3D GAN training is limiting in several ways: it does not allow for topology changes, it does not prevent solid parts like teeth or eyeglasses from being stretched, and the deformation quality degrades for points far from the surface. In general, using the surface of a parametric mesh to guide an volume may not be the optimal choice for more complex volumetric scenes. More advanced deformation methods, perhaps including kinematic constraints, could be an interesting direction of future research.

**Ethical considerations.**    GANs could be misused for generating edited imagery of real people. Such misuse of image synthesis techniques poses a societal threat, and we do not condone using our work with the intent of spreading disinformation. We also recognize a potential lack of diversity in our results, stemming from implicit biases of the datasets we process.

**Conclusion.** Our work takes important steps towards photorealistic 3D-aware image synthesis of articulated human bodies and faces with applications in visual effects, virtual or augmented reality, and teleconferencing among others.

## Acknowledgments and Disclosure of Funding

Alexander W. Bergman was supported by a Stanford Graduate Fellowship. Gordon Wetzstein was supported by NSF Award 1839974, Samsung, Stanford HAI, and a PECASE from the ARO. We thank Connor Lin for helping standardize comparisons to [130]. We thank Atsuhiro Noguchi and the authors of [118] for helping in standardization of the comparisons of our methods.

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
