# Generative Neural Articulated Radiance Fields
# –Supplementary Document–

**Alexander W. Bergman**\*
Stanford University
awb@stanford.edu

**Petr Kellnhofer**\*
TU Delft
p.kellnhofer@tudelft.nl

**Wang Yifan**\*
Stanford University
yifan.wang@stanford.edu

**Eric R. Chan**\*
Stanford University
erchan@stanford.edu

**David B. Lindell**
University of Toronto
Vector Institute
lindell@cs.toronto.edu

**Gordon Wetzstein**
Stanford University
gordonwz@stanford.edu

computationalimaging.org/publications/gnarf/

## Contents

---

\*Equal contribution

36th Conference on Neural Information Processing Systems (NeurIPS 2022).

# 1 Implementation Details

Applications of our method to human bodies and faces share the same framework, but differ in a few implementation details. For clarity, in the following section we describe the implementation details for GNARF applied to human bodies and outline the differences for faces in Sec. 2.3.

## 1.1 Generator and tri-plane representation

We use the generator architecture from EG3D [1], which is built on top of the public StyleGAN2 [2] architecture located at `https://github.com/NVlabs/stylegan3` (this StyleGAN3 repository contains backward compatibility for StyleGAN2). The generator is composed of four components: a mapping network, a convolutional backbone, an MLP decoder, and a convolutional super-resolution module.

The generator is conditioned on a 512-dimensional Gaussian noise input using a two-layer mapping network of 512 hidden units. We do not condition the generator on either camera pose or body pose. The mapping network produces a 512-dimensional latent code. This latent code modulates the layers of a StyleGAN2-based convolutional backbone, which produces a 96-channel $256 \times 256$ feature image. This is reshaped into three axis-aligned tri-planes, each of shape $256 \times 256 \times 32$. This architecture is trained from scratch rather than using a pre-trained StyleGAN2 network.

The MLP decoder which operates on top of sampled plane features consists of a single hidden layer of 64 units. The decoder maps the 32-dimensional sampled plane feature to a 33-channel feature consisting of a scalar density and 32-dimensional feature. These are integrated per the volume rendering equation (Eq. 3 in the main paper) to obtain a $64 \times 64 \times 32$ feature image, where $64$ is the spatial resolution and $32$ is the number of channels.

As in EG3D, a separate super-resolution module (implemented as CNN) up-samples and converts the feature images to the final RGB output. The final resolution of the output is $128^2$ for SURREAL and $256^2$ for AIST++ respectively. As in EG3D, this module is implemented with two StyleGAN2 convolutional blocks, with channel depth of 128 and 256 respectively.

## 1.2 Deformation and volume rendering

The SF deformation is performed using a simplified version of the SMPL mesh. As described in the main text, this simplified version is obtained using Quadratic Error Metric Decimation [3] in the Open3D library [4] to reduce SMPL from 6890 vertices and 13,776 faces to 690 vertices and 1376 faces.

We perform volume rendering in the canonical space with 64 uniformly-spaced samples plus 64 additional samples based on importance sampling [5] per ray. Additionally, we sample rays only inside an expanded version of the simplified SMPL mesh. The mesh is expanded using a growth offset parameter $g$, which controls the new position of a vertex $v$ with vertex normal $n$, by moving $v$ to $\hat{v} = v + gn$. We use $g = 0.05$ during training.

As described in the main paper, deformation methods such as Mean Value Coordinates (MVC) [6] and skinning introduce large artifacts when the template mesh (accidentally) intersects itself or comes very close to doing so. This happens often in practice, even with the unmodified SMPL mesh, due to imperfect SMPL parameter estimation or a subject actually touching somewhere on their body.

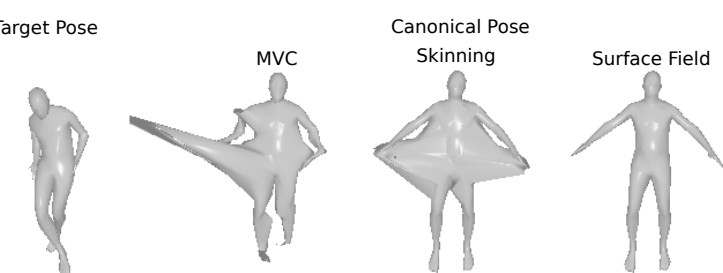

Figure 1: Deforming the vertices of the target pose on the left into the canonical pose results in artifacts for the MVC and skinning deformation method, while the surface field deformation method results in a perfect deformation by construction.

This is shown in Fig. 1: the hands coming close to the body result in large deformation artifacts in the canonical space for MVC and skinning. In the case of MVC, this is mainly attributed to MVC being non-local - a point has non-zero weight w.r.t. *all* the vertices in the driving mesh. This is worsened by the grid approximation mentioned in the main paper (this is necessary to enable feasible training time using MVC): the grid approximation to MVC does not guarantee that surface points remain on-surface after deformation (unlike the full MVC computation) since the rapid change in MVC weights near the surface cannot be sufficiently captured in feasible grid resolution. Similarly, in the skinning method, when parts of the driving mesh get close together, individual vertices may become closer to a different bone than the one which they are a part of. This causes them to deform to incorrect places in the canonical space. This artifacts are significantly mitigated when using the proposed surface field (SF) deformation, as SF is very local (in contrast to MVC) and the lookup of the closest triangle is less error-prone than of the closest bone.

## 1.3 Discriminator

**Dual discrimination.**    Similarly to EG3D, we use dual discrimination to ensure consistency between the raw neural rendering and the final super-resolved output. As in EG3D, we concatenate a resized copy of the raw neural rendering to the super-resolved input to form a 6-channel discriminator input tensor. This raw neural rendering consists of the first three channels of the rendered feature image.

**Discriminator pose conditioning.**    As in EG3D, we condition the discriminator on the camera pose via a mapping network that modulates the layers of the discriminator. Unlike EG3D, we additionally condition the discriminator on the expected body pose / facial expression by concatenating the body/facial pose parameters (SMPL or FLAME parameters) to the camera parameters as input to the mapping network.

By conditioning on body / facial poses, we give the discriminator the ability to ensure that the applied deformation matches the specified pose. Empirically, we found that corrupting the poses with 1 standard deviation of Gaussian noise before passing them as input to the discriminator aided training convergence. We hypothesize that in the absence of noise, the discriminator was able to overfit on the specific poses and cameras of the ground truth dataset, which destabilized training.

Note that unlike EG3D, which conditions the generator on the camera parameters for training with FFHQ *(modeling pose-correlated attributes)* we do not condition the generator with camera pose. We also do not condition the generator with body pose or facial expression. This is done in order to ensure that the generator learns to generate a body/face in the canonical space which is robust to custom deformations, rather than one which is specific for a warping or camera viewpoint.

## 1.4 Training

Many of our training hyperparameters are adopted from those of EG3D and StyleGAN2: generator learning rate (0.0025), discriminator learning rate (0.002), batch size (32), blurring images (GT and generated) over the first 200K iterations, and R1 regularization [7]. As recommended, the gamma parameter of R1 regularization is tuned according to the dataset: FFHQ: $\gamma = 1$; SURREAL: $\gamma = 1$; AIST $\gamma = 4$.

We use 8 Tesla V100 GPUs, training each model for roughly 2 days.

## 2 Additional Results and Evaluation Details

### 2.1 Single-scene overfitting.

**Data pre-processing.**    The single-scene overfitting experiment is conducted using the sequence *gBR_sBM_cAll_d04_mBR0_ch01* from the AIST++ dataset [8]. We extract all 719 frames sampled at 60 Hz from each of the 8 available cameras and we crop the human body using the same procedure as in the GAN training. We skip the camera number 4 because the annotation in the dataset does not match the video. We hold out the cameras number 2 and 7 for testing and we train our models using 30 frames uniformly sampled from the remaining six cameras.

**Model.** We use the same triplane representation as in our GAN experiments with a few modifications to avoid overfitting to the sparse training data. First, we limit the capacity of the decoder network by reducing the latent space size from 64 to 32 and by reducing the resolution of the triplanes from 256 to 128. Next, we increase the number of samples per ray to 128 and disable the second stage of importance sampling. Finally, unlike in the GAN setup, we do not remove background from the training images. Instead, we train a representation of the entire scene as is common in other overfitting papers [9]. To avoid mixing of the static background and dynamic foreground in the neural representation and to allow for efficient sampling of both regions with different depth ranges, we include a separate identical triplane representation for the background. We use the accumulated optical density from the foreground network to alpha-blend the foreground image over the background image. No ground-truth foreground masks are used during the training.

We utilize the same model for deformation function $D$ implemented using mesh skinning, Surface Field and MVC. For skinning and Surface Field, we compute the transformations on-the-fly. However, this is not feasible for the relatively slow MVC computation. Therefore, we precompute the MVC transformations for $16{\times}16{\times}16$ points uniformly sampled withing a bounding cube of the human body for all training poses, and we sample them during training using a trilinear interpolation.

We train our model for 500 000 steps with Adam optimizer [10] and step size of 0.002 and we use L2 loss to supervise the training at $128{\times}128$ resolution with batch size of 3 images on Nvidia RTX3090 graphical processing unit.

**Evaluation.** We rely on well known image metrics to compare performance of individual warping methods. Since the goal is to evaluate efficacy in compensating human body motion and not capacity for learning the background scenery, we use foreground masks for computing the image metrics. To this goal, we compute human body masks using a pre-trained image segmentation model [11]. Then, we filter out the background pixels for the PSNR metric. For the structural metrics of SSIM and LPIPS [12], we set the background pixels to zero in both the predicted and ground-truth images.

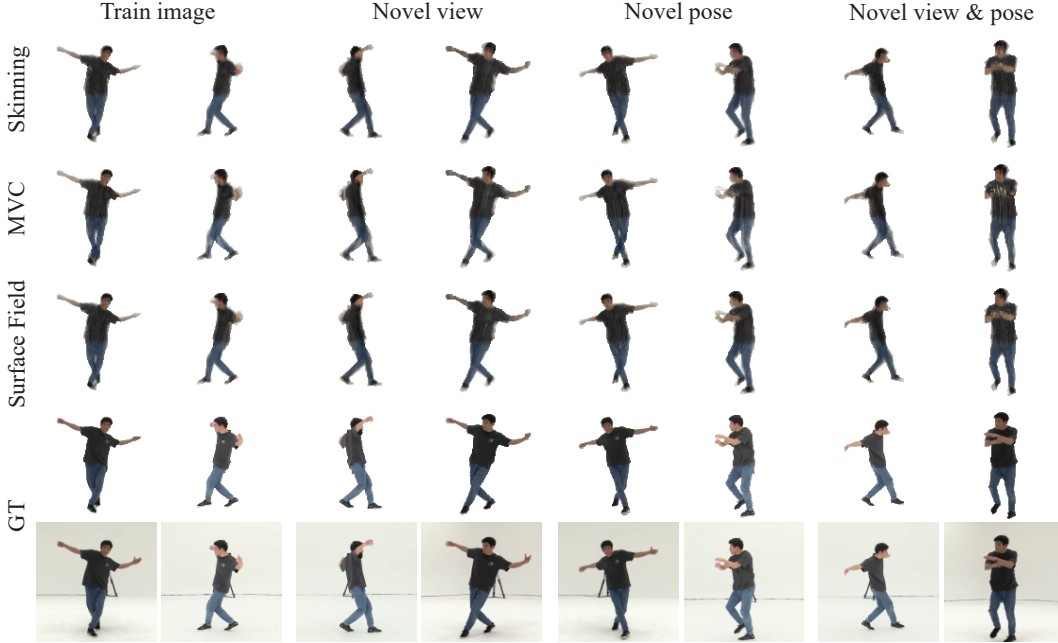

Figure 2: Qualitative comparison of results from our single-scene overfitting experiment. The figure presents the original training views and poses, interpolation of novel views, interpolation of novel poses and interpolation of novel poses under novel views. All images except the last row are presented with the same foreground masks as used for the metric evaluation.

## 2.2 Human body generation and animation

**SURREAL data pre-processing.** For training, we use the official SURREAL training split, and extract the first frame from each video. We square-crop each frame from head-to-toe using the ground truth segmentation mask, and resize the image to $128 \times 128$. The backgrounds are set to be black. We use the provided camera poses and intrinsics from the SURREAL dataset for training. Images for which the SMPL mesh scale is not consistent with the image size are filtered, leaving 35,332 total images for training.

The SURREAL dataset provides ground truth SMPL parameters for each frame, which we use. We compute the mean SMPL parameters across the frames in the dataset in order to assign the canonical pose, such that it is close to each of the target SMPL poses.

**AIST++ data pre-processing.** For training, we extract 30 frames uniformly sampled from each video in the AIST++ dataset. We then filter out frames whose effective camera distance to a normalized SMPL model is above a threshold or the projected human bounding box is partially outside of the image as a form of heuristic detecting poorly estimated SMPL parameters. We square-crop these frames at $600 \times 600$ resolution centered at the pelvis joint of the SMPL mesh, and resize each image to $256 \times 256$. Since ground truth masks are not provided for this dataset, we run an off-the-shelf segmentation model [11] to remove backgrounds. This stabilizes the GAN training, as the GAN no longer has to attempt to model a 3D-consistent background.

The AIST++ dataset also provides ground truth camera and body pose parameters. We move the translation of the SMPL mesh into the camera extrinsics, and simulate the scaling of the SMPL mesh by either moving the camera further back or closer. We additionally rescale all meshes and camera parameters such that the mean distance from camera to mesh is 1.7. Similarly to SURREAL, we compute the mean SMPL parameters across the selected training frames to assign the canonical pose.

**Evaluation.** We use the FID metric [13] for evaluation of the quality of generated images. This metric compares the distribution of intermediate features extracted from an inception network run on both generated and ground truth images. FID (10k) refers to the evaluation consistent with [14], which compares the distribution of 10,000 generated images with 10,000 randomly sampled ground truth images. FID (50k) compares the distribution of 50,000 generated images with the entire training dataset, giving a better estimate of the true distance. For EG3D, FID is run on the generated outputs with no warping. For the EG3D + warping baseline, the generated results from EG3D (not in the canonical pose) are warped by estimating the pose of the generated body and using this as the canonical pose. The FID is then applied to images generated in this fashion. For GNARF, we simply apply the FID to generated images with our deformation method.

In order to measure deformation accuracy for our method and the EG3D + warping method, we use the PCKh@0.5 metric. The use of this metric was inspired by the concurrent work [14]. After correspondence with the authors of this work, we have standardized our evaluation of this metric in order to compare reported values in the main paper table. Details of the evaluation method have a large effect on the magnitude of the PCKh@0.5 numbers reported, but describe the correct trend within a consistent evaluation standardization. For both the AIST++ and SURREAL dataset, we compute the GT keypoints by running a off-the-shelf body keypoint estimator [15] trained on MPII [16] (publicly available on the MMPose Project [17]) on each GT image. We then generate an image and deform the generated result with the body-pose parameters corresponding to this GT image, and render the generated result from the same camera position. The keypoint estimator is then run on this generated image, and the keypoints detected are compared in 2D. We discard keypoints that the keypoint estimator is not confident on in order to factor out estimator error, and only compare confident keypoints. We determine the head size (interocular distance) using the detected keypoints from the GT image.

## 2.3 Human face generation and editing

Here, we outline the implementation differences in the human body application.

**Deformation.** We use the FLAME head model to drive the deformation. The original template FLAME [18] template model has 5023 vertices and 9976 faces. This mesh contains 3 unconnected

parts, modeling the base face and the two eyeballs respectively. Since there is no suitable method to accurately extract the FLAME parameters related to eye movements from training images, we remove the eyeball parts. In original template, the neck and mouth are modeled with holes. We find having holes slightly degrades the deformation quality for points around the hole area, likely because worse point-to-triangle correspondence. Furthermore, we find the small triangles can lead to numeric instabilities when e.g. computing the barycentric coordinates for deformation, therefore we decimate the mesh as described in the main paper which makes the triangle sizes more uniform and also speeds up the SF deformation. The resulting mesh template, shown in Fig. 3 contains $1,252$ vertices and $2,500$ triangles.

The default FLAME head model is in a different scale and origin as the head generated by the pretrained EG3D model. To ensure meaningful warping in the transfer learning, we rescale by $2.6$ and fix the root joint at $[-0.0013, -0.1344, -0.0390]^\mathsf{T}$, which we determined by visually aligning the heads from FLAME and a pretrained EG3D.

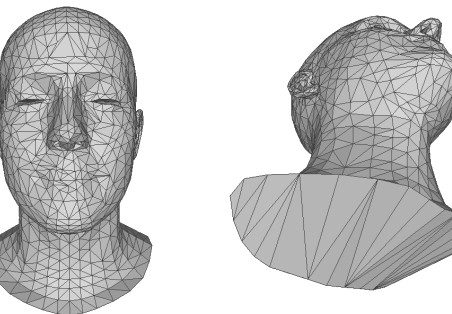

Figure 3: Processed FLAME model for training GNARF on FFHQ.

**Generator pose conditioning.** Unlike for the body, we use camera pose conditioning for the generator as proposed by EG3D. EG3D found that by using swapping regularization, camera pose conditioning will not negatively impact 3D consistency but rather improve generation quality. We found this true in our experiments for faces.

However, same as the body experiment, we do not provide the generator with any information related to the deformation (i.e. expression, shape and jaw rotation). As explained before, this is crucial for generating consistent canonical faces.

**Volume Rendering.** Unlike for bodies, the final resolution of the output is $512 \times 512$, consistent with the original EG3D.

**Dataset preprocessing.** Our data preparation is based on the original EG3D [1]. For each training image, we fix the camera intrinsics and estimate the camera extrinsics assuming that the head is inside a unit-length bounding box, front-facing and at a fixed position. Additionally, we use DECA [19] to estimate the expression, pose (only the jaw rotation, since we assume that the head is front-facing) and the shape parameters of the FLAME models. Note that, originally, DECA also outputs the camera parameters. However it uses an orthographic camera model, which is not directly transferable to the camera used in the pretrained EG3D. We therefore use the camera pose estimation from EG3D's data preprocessing procedure.