# OpenReview forum: "Generative Neural Articulated Radiance Fields"
_NeurIPS.cc/2022/Conference — NeurIPS 2022 Accept_

### Official Review · Reviewer_UgRe · 2022-07-08

**Rating:** 7
**Confidence:** 4
**Soundness:** 3 good
**Presentation:** 3 good
**Contribution:** 3 good

**Summary:**

This paper proposes a method for generative neural articulated radiance fields. The main strategy is to generate radiance fields in a canonical pose and warp them using an explicit deformation field into a desired body pose or facial expression. The main technical contribution of this work is to combine the recently proposed tri-plane feature volume representation with an explicit feature volume deformation which is guided by a template shape. High-quality results on human body and faces are demonstrated in the paper.

**Questions:**

I don't have questions.

**Limitations:**

As mentioned in the related work, HeadNeRF is related to this work, and the authors point out that HeadNeRF needs to acquire training images of the same person performing various expressions in different lighting conditions. However, HeadNeRF can explicitly control properties like identity and expressions. To my knowledge, the proposed method can not achieve disentangled representation.

**Strengths And Weaknesses:**

This paper proposes a practical 3D-aware GAN framework for the generation of editable radiance fields of human bodies. The main strength of this paper is to combine several existing strategies like tri-plane feature volume representation, StyleGAN2 generator, feature volume deformation, neural volume rendering, image super-resolution, etc. Although each component has been proposed before, integration them together to generate satisfying results is not easy and could be used for related applications.

On the other side, the weakness is the lack of technical contribution as each component is borrowed from other papers. Meanwhile, some component can be replaced with more suitable strategy. For example, based on our implementation and the experimental results reported in the paper, tri-plane representation still has limited representation ability, which can be replaced by other representations to achieve better performance.

---

> ### Author Response · Authors · 2022-07-30
> **Reviewer UgRe response**
>
> We thank Reviewer UgRe for their time spent reviewing and commenting on our work. We appreciate the note that integrating advanced radiance field implementation architectures, generative models, and articulation is not trivial and is an important contribution for future applications.
>
> **Lack of technical contributions**
>
> While it is true that 3D-aware generative models and articulated radiance fields, and their implementation details such as the tri-plane architecture, have been explored in the past, no other work attempts to combine them and enable our application: generative, controllable 3D models of objects. We demonstrate that this applies to not only human faces, but also bodies, which have not been previously explored in the context of 3D-aware generative models with radiance fields. Moreover, we show that the trivial combination of these components, the EG3D + re-warping baseline, does not perform nearly as well as the proposed GNARF architecture, demonstrating that the combined architecture is an important contribution in of itself.
>
> While it is possible that individual components of this combination can be improved, such architectural improvements would be complementary to our work. These improvements are also orthogonal to our core technical contribution: enabling articulated generative modeling by factoring the generator into canonical pose generation and deformation applied to neural radiance fields.
>
> **HeadNeRF comparison**
>
> We note that HeadNeRF is similar in input and application in that it generates 3D representations of heads conditioned on both identity and expressions (along with other attributes, such as albedo and illumination, which GNARF does not factorize), and is able to generate very high-quality results. Since GNARF does not factorize lighting and albedo, it cannot render the same identity under varying lighting conditions, which we will note as a limitation in our paper.
>
> However, for training data, HeadNeRF requires training images which hold identity constant while varying expression and other parameters, which is not required by GNARF. This allows GNARF to be trained on single-view datasets.
>
> Both GNARF and HeadNeRF are in fact able to control identity and expression independently (disentangled). In GNARF, the identity variation is modeled by the latent code, which generates the body or face in a canonical pose. The target SMPL mesh and deformation method model the expression (or, in the case of bodies, pose) variation. This explicit separation during training encourages the model to disentangle these two parameters. This is supported in our experiments, where walking through the latent space does not affect the body pose, and similarly, editing the body pose does not affect the body identity (see supplemental video). HeadNeRF is able to do the same, along with other important conditioning information, but is limited by the requirement on input data. We will be sure to edit the paper to make this comparison more clear, highlighting the differences and similarities between the methods, and the difference in requirements and what applications they enable.

---

### Official Review · Reviewer_H6JG · 2022-07-11

**Rating:** 7
**Confidence:** 5
**Soundness:** 3 good
**Presentation:** 3 good
**Contribution:** 3 good

**Summary:**

This paper proposes a technique called GNARF for controllable 3D aware generation of human bodies and faces with different articulations and expressions. The authors build on the prior EG3D generative model and extend it along two dimensions -- to entire 3D bodies and for more explicit control and disentangling of base identity faces and their expressions. The authors' main idea is to introduce an explicit surface deformation module into the generative model, which deforms a canonical triplane representation. The authors parametrize this surface deformation via parametric mesh SMPL or FLAME models. They compare their surface deformation approach against MVC and blend-skinning and also their generative models for body and face against SOTA existing generative models and show improvements in quality.

**Questions:**

From Table 2, it is evident that the baseline EG3D (no warping) approach has FID scores nearly as good as or better than GNARF, meaning that it is able to generate highly realistic posed images. However in line 244 the authors claim that for EG3D (no warping) poor results for reposed images are produced "since it is difficult to accurately estimate the SMPL mesh from the generated images". If the image quality of the posed images is good as per the FID scores, what limits good SMPL mesh fitting, and reposing of the EG3D generated images?

How do the authors propose to handle cases such as subject wearing long skirts or dresses, which are likely to not fit well to the two-legged tight-fit model of SMPL?

**Limitations:**

The authors have adequately and honestly discussed the limitations of their proposed method.

**Strengths And Weaknesses:**

- Originality: The work is original along many dimensions. It extends 3D-aware generative models to body poses, which has not been considered in prior work (other than in [115], which is concurrent work). It is also novel in introducing the explicit surface deformation model  into the EG3D framework, making it more controllable. However, it builds heavily on EG3D. Nevertheless, in my opinion this work introduces sufficient novelty over the existing works to warrant publication.

- Quality: The work is technically sound; the method appropriate and its steps logical; and most claims (a few exceptions noted below) are well supported via experiments. To the authors' credit they have also gone above and beyond the submission requirements and compared against several unpublished concurrent works [115, 127] and shown the superior performance of their approach. The authors have discussed the limitations of their method honestly and in much detail.

- Clarity: The material is clearly presented and easily understandable. Many implementation details are included in the supplementary material and the authors have promised to release their code upon acceptance.

- Significance:
- The dominant approach for modeling humans (bodies and faces) and animals has been via parametric mesh models for several decades now. The advent of 3D aware GANS is heralding in a new era of GAN-based generative models, which have the promise to provide much higher images quality and photo-realism, while not being limited only to the surfaces (face and body) modeled by the mesh models. 3D aware-GAN models can also model hair, eyeglasses and clothing details beyond meshes. However, the question of how best to model body articulation/surface deformation caused by expressions/body articulation into the context of 3D-aware GANS is an important open one. This work provides an important step towards addressing this latter problem. That said the proposed solution's falling back on parametric mesh models to model the surface deformation is perhaps not the most elegant solution, as the authors acknowledge themselves as well.
- The authors' comparisons of the different types of 3D mesh-based deformations (MVC, blend-skinning and surface-based) is also quite interesting and insightful.

---

> ### Author Response · Authors · 2022-07-30
> **Reviewer H6JG response**
>
> We thank Reviewer H6JG for reviewing and providing a thoughtful and extensive analysis of our work. We appreciate the positive comments recognizing the originality of extending 3D-aware generative models to controllable bodies and clarity of presentation. We also appreciate that the work is viewed as an important step forward in addressing the significant problem of integrating articulation into 3D-aware generative models.
>
> **Using a parametric mesh to drive surface deformations**
>
> We acknowledge that the use of the parametric mesh surface to control a generated radiance volume may not be optimal in every context. We also mention in the future work that further improvements in deformation could be leveraged in this same overall pipeline, such as learning the deformation method. We will make this more clear in the manuscript. However, we believe that the mesh-based modeling and deformation is intuitive and enables interpretable control, and naturally maps to traditional computer graphics and mesh manipulation.
>
> **Limitations of the two-legged SMPL model**
>
> Although the deformation method is guided by the two-legged SMPL model, the surface-field deformation is still defined over the entire volume, by warping 3D points with their closest surface point. This allows the deformation method to have robustness to cases where the object modeled by the radiance field does not exactly fit the template, for example the hat on the person in the second row of figure 1. Explicitly modeling the deformation of more complex clothes or accessories which may not move intuitively with the surface of the SMPL mesh could be similarly addressed in the future with more complex deformation functions integrating clothes simulation models from computer graphics.
>
> **Limitation for re-posing EG3D models**
>
> As mentioned the EG3D generated images are of a similar quality as those generated by GNARF. However, estimating the SMPL mesh from any image is not a solved problem. We use the state of the art method, SPIN [31], for estimating SMPL parameters from images rendered from the generated EG3D models. However, this method still has bias based on the data that it was trained on: less diverse pose distribution than the synthetic or AIST++ dataset since in-the-wild poses are often biased towards a neutral position, and more complex backgrounds in in-the-wild images. We view these limitations of the estimation of these parameters as a drawback of using the EG3D baseline, as an additional step must be injected into the system, possibly resulting in more error, while the GNARF generator automatically generates objects in a canonical pose. Additionally, incorrect geometry that sometimes exists in standard EG3D (floating but indiscernible content in the background) could turn into foreground occluders as a result of a naive post-process deformation. Generating these geometric artifacts is not penalized in the EG3D training pipeline, but is penalized in our end-to-end training procedure.

---

### Official Review · Reviewer_qMy8 · 2022-07-11

**Rating:** 6
**Confidence:** 5
**Soundness:** 3 good
**Presentation:** 3 good
**Contribution:** 3 good

**Summary:**

This paper proposes Generative Neural Articulated Radiance Fields, named GNARF, serving as the 3D representation of 3D-aware generator to handle the datasets with more deformation, such as human-body datasets. The non-rigid motion is represented by the surface deformation derived from the source and target meshes, which helps the tri-plane focus on learning the canonical feature volume. The experiments give a comprehensive study of GNARF on several human datasets, including ATISS++, SURREAL, as well as FFHQ. The experimental results demonstrate the GNARF can achieve better image quality and editability compared with several baselines.

**Questions:**

1. As shown in this paper, SF is derived from the target and canonical template meshes. Does it mean that each batch also needs to sample the target SMPL models besides the latent code? If I am right, it is not fair to compare with the EG-3D baseline w.o. any 3D mesh information.
2. It is not convincing that the authors remove the background to stabilize the training process. Why not use a separate background model? StyleNeRF has demonstrated that it can work even on the CompCars dataset with large background variations.
3. Since the tri-plane is required to learn the canonical feature volume, why is the latent code still injected into the tri-plane generator network? Why not only add the stochastics to the shallow MLP because the canonical pose across different instances is similar.

**Ethics Review Area:**

["I don’t know"]

**Limitations:**

My concerns about this paper is the synthesis quality. Although it has improved on the baseline, it still has a large gap to real-world applications like avatar animation.
Besides, the mouth of the synthesized faces in Fig.6  and the demo video are not realistic. I guess that the Flame model does not model the teeth. Does it mean that the GNARF is very dependent on the template meshes? When transferred to a new dataset like LSUN cats which does not have any ground truth template meshes, how does GNARF perform?
I think this paper is interesting, and I tend to accept this paper (raise the scores) if the authors can address my concerns.

**Strengths And Weaknesses:**

This paper aims to generate 3D humans from a posed 2D image collection, which is rarely explored in the literature. Compared to prior arts of 3D-GANs, the paper pays more attention to the non-rigid human bodies instead of the rigid objects like faces and cars. It is valuable to solve this challenging problem. Although this task is very difficult, the synthesized results are not very good. The details of the human body, such as face and clothing,  are of a low quality. Besides, I also have some confusion about the proposed techniques, which will be presented in the following section.

---

> ### Author Response · Authors · 2022-07-30
> **Reviewer qMy8 response**
>
> We thank reviewer qMy8 for spending time on reviewing and providing insightful comments regarding our paper. We appreciate the acknowledgment of the novelty and significance of the method in approaching a problem which is rarely explored in literature but challenging and impactful.
>
> **Quality of the Results**
>
> We acknowledge in the limitations section that there is room for improvement on the fine details of the body and believe this to be an important direction for future work. We emphasize that this is the first work which leverages neural radiance fields as a 3D representation in a generative framework for bodies, and that our work outperforms logical extensions of prior work and concurrent baselines in both generation quality and control. Additionally, existing datasets which provide a distribution of SMPL pose parameters associated with images are limited in diversity (AIST++ only has 16 subjects) and fine detail quality (SURREAL is synthetically generated at low resolution). With improved datasets, we expect that our method can generate higher quality and more diverse human body results.
>
> **Comparison of EG3D and GNARF (Q1)**
>
> It is correct that each batch samples the target SMPL model along with the latent code. Both our method and the baseline (EG3D + re-warping) sample the latent code and target SMPL pose, and perform a generation followed by warping operation in order to generate a 3D model in the target pose. These two methods receive the same input information, and are compared via the same metrics: FID, which evaluates the diversity and realism of the distribution of generated images, and PCKh@0.5, which evaluates the accuracy of the generated image to the target pose. This comparison fairly analyzes the quality of generated images and accuracy to the target pose.
>
> The base EG3D only takes in a latent code without target pose parameters. As such, we do not compare EG3D to our method in terms of PCKh@0.5, as we don’t expect the generated EG3D bodies to match the target pose and this would not be a fair comparison. We only compare GNARF to EG3D in FID.
>
> GNARF explicitly factorizes the conditioning into a latent code controlling identity and SMPL parameters controlling pose, while EG3D combines both identity and pose into the learned latent space. However, both attempt to model the same distribution - the identities and poses of humans in the dataset. FID compares how GNARF and EG3D model this distribution, thus ensuring that this is a fair comparison, since it assesses the generated diversity and quality of rendered images of the 3D models rather than the ability to match a specific target pose given as input.
>
> **Background modeling (Q2)**
>
> StyleNeRF, EG3D, and other GAN works are able to model backgrounds, but these methods do not attempt to generate a radiance field which can be animated. The addition of animation makes the problem more challenging because the deformation method must apply logically to both the object and the background independently. In the single scene overfitting experiments (supplementary sect. 2.1), we introduce a separate tri-plane representation to model the background to ensure consistency across the animated frames. However, our main contribution for the generative model is to generate realistic animatable 3D representations of bodies and faces. We will note this in the limitations and future directions section of our manuscript.
>
> **Latent code and generator architecture (Q3)**
>
> The latent code is injected into the tri-plane feature generator, as the latent code controls the identity of the generated object but not the pose. The pose animation is controlled by the SMPL parameters, which are not input in any way to the generator, since the generator is only tasked with generating the canonical pose.
>
> Choosing where to input the latent code (StyleGAN2 generator or the shallow decoding MLP) is an architectural change to the generator which does not change the overall input or interpretation. While all identities are generated in the same (canonical) pose, the appearance variation due to identity is enough that only adding the conditioning into the shallow MLP may not provide enough capacity to model the dataset diversity. We will edit the paper to be more clear about this design decision, and in general about the various parameters and their interpretation.
>
> **Dependence on template meshes (Limitation)**
>
> GNARF is not highly dependent on the accuracy of the template mesh, as our template meshes have been reduced from the full FLAME and SMPL model significantly without a decrease in quality. However, using a template mesh to drive pose animation provides an interpretable articulation method, and thus we apply our method to classes where this template exists. When extending this method to classes which do not have a template mesh, such as the LSUN cats, the interpretability of the pose control is not clear as it’s not known which joints cats move their faces around.

---

### Meta-Review · Area_Chair_H8Bv · 2022-08-27

**Recommendation:** Accept
**Confidence:** Certain

**Metareview:**

The reviewers all recognize the quality of the work, particularly technical soundness and quality of the experimental setting and there is a clear consensus for acceptance. I ask the authors to address the reviewers concerns, particularly clear up any confusion in the manuscript and better analysis of the synthesis results.


**Award:**

No

---

### Decision · Program_Chairs · 2022-09-14

Accept